# Detecting Early Ocular Choroidal Melanoma Using Ultrasound Localization Microscopy

**DOI:** 10.3390/bioengineering10040428

**Published:** 2023-03-28

**Authors:** Biao Quan, Xiangdong Liu, Shuang Zhao, Xiang Chen, Xuan Zhang, Zeyu Chen

**Affiliations:** 1The College of Mechanical and Electrical Engineering, Central South University, Changsha 410083, China; 2The Department of Dermatology, Xiangya Hospital, Central South University, Changsha 410008, China; 3The Department of Ophthalmology, Xiangya Hospital, Central South University, Changsha 410008, China

**Keywords:** deconvolution, early detection, ocular choroidal melanoma, three-frame difference, ultrasound localization microscopy

## Abstract

Ocular choroidal melanoma (OCM) is the most common ocular primary malignant tumor in adults, and there is an increasing emphasis on its early detection and treatment worldwide. The main obstacle in early detection of OCM is its overlapping clinical features with benign choroidal nevus. Thus, we propose ultrasound localization microscopy (ULM) based on the image deconvolution algorithm to assist the diagnosis of small OCM in early stages. Furthermore, we develop ultrasound (US) plane wave imaging based on three-frame difference algorithm to guide the placement of the probe on the field of view. A high-frequency Verasonics Vantage system and an L22-14v linear array transducer were used to perform experiments on both custom-made modules in vitro and a SD rat with ocular choroidal melanoma in vivo. The results demonstrate that our proposed deconvolution method implement more robust microbubble (MB) localization, reconstruction of microvasculature network in a finer grid and more precise flow velocity estimation. The excellent performance of US plane wave imaging was successfully validated on the flow phantom and in an in vivo OCM model. In the future, the super-resolution ULM, a critical complementary imaging modality, can provide doctors with conclusive suggestions for early diagnosis of OCM, which is significant for the treatment and prognosis of patients.

## 1. Introduction

Ocular melanomas often originate from the uveal with rich blood supply, of which 85% occur in the choroid and some occur in the ciliary body (10%) and iris (5%) [1,2]. Ocular choroidal melanoma (OCM) is the most common primary intraocular tumor in adults, which is characterized by high malignancy, easy invasion and metastasis, and poor clinical prognosis [3,4]. It is estimated that approximately 6 in 1 million white people suffer from OCM, especially those aged 50 to 80 [5,6,7]. In addition, the 5-year mortality rate for patients with small tumors (<3 mm thickness) is 16%; for medium tumors (3–8 mm thickness), it is 32%; and for large tumors (>8 mm thickness), it is 53% [8]. The risk of metastasis gradually raises with increasing tumor thickness, resulting in a relatively poor prognosis. Small OCM, located in the periphery of the eye, may have no clinical symptoms, or may only cause slight vision loss in patients [9]. In the later stage, with the increase in tumor size, local invasion or distant metastasis, the visual function of patients will be seriously damaged or even life-threatening [10]. Therefore, the detection of OCM early in its natural course and microcirculation evaluation, such as blood flow velocity and vascular morphology, are of excellent clinical value to prevent distant metastasis of melanomas and improve the prognosis of patients.

B-mode ultrasound is a widely accepted clinical imaging modality for the detection and diagnosis of OCM. It offers superior real-time, non-invasive, safe and economic performance in contrast with other approaches. However, gray-scale ultrasound cannot render the key parameters of hemodynamics and suffers from difficulties with early detection of OCM relating to its clinical similarity to benign choroidal nevus [11,12]. Despite the emergence of color Doppler imaging (CDI) which can directly display the blood supply of the lesions to counter the deficiency of ultrasound in blood flow speed images, a crucial constraint of CDI is its insufficient capacity in assessing small vessel signals and slow blood flow velocity in the eye [13,14].

Indocyanine Green Angiography (ICGA) plays a critical role in visualizing choroidal circulation and hemodynamic information. On the other hand, it depends on a superb transparent refractive medium and is contraindicated in patients with iodine or shellfish allergies, liver disorder, and end-stage renal disease [15,16].

In Magnetic Resonance Imaging, melanomas existing paramagnetic properties show high signal intensity on T1 weighted images and mildly low signal intensity on T2 weighted images. However, this modality of imaging lacks the ability to propose hemodynamic information and rarely distinguishes subacute early hematoma [17,18].

Super-resolution ultrasound (US) microvessel imaging, also known as ultrasound localization microscopy (ULM) shows promising clinical benefits for the detection and evaluation of ocular illness, which combines deep penetration and super-resolution below the diffraction limit by localizing the signal of spatially isolated microbubble (MB) position [19,20]. Compared with conventional plane-wave US imaging exerting unfocused and single-angle ultrasonic plane waves, ULM based on coherent plane-wave compounding imaging performs high-quality images and ultrafast frame rates imaging capabilities after coherently superimposing the received echo ultrasound signals. The ability of ULM to accomplish microvessel imaging in vivo has been demonstrated [21,22,23,24]. ULM requires precise MB localization for post-processing steps to reconstruct super-resolution microvessel images and struggles to acquire abundant MB signals from the region of interest (ROI) [25]. These inferior performances extremely limit the development of ULM in the diagnosis of OCM.

To fill these gaps, in this paper, we propose a novel method for obtaining the kernel of an image deconvolution algorithm, present an ultrafast plane wave imaging which capturing MBs in ROI based on a three-frame difference algorithm and fulfill the detection of OCM in vivo rat with ULM. The ultrafast plane wave imaging is not only used to remove tissue signals and extract MB signals, but also display the location of MBs in a microvessel in real-time. Our study enables doctors to more conveniently detect whether there are blood flow signals in the ROI and presents superior consequences for the signal-to-noise ratio (SNR) of super-resolution microvessel images and blood flow velocity calculations. In the future, ULM may be applied in the clinical evaluation of morphological and hemodynamic characteristics in OCM, which is beneficial for diagnosis, treatment decision making and prognosis improvement of patients.

## 2. Materials and Methods

### 2.1. System Setup

Figure 1 shows our experimental setup for microvessel imaging of the OCM. A L22-14v linear array transducer (Verasonics Inc, Kirkland, WA, USA) with a center frequency at 18 MHz connected to a high-frequency Verasonics Vantage system (Vantage 256, Verasonics Inc., Kirkland, WA, USA) was used to multi-channel transmit waveform generation, analog signal amplification and filtering, and digital signal processing. In order to facilitate the ROI imaging, the transducer was stabilized on a three-axis stepper motorized stage with a minimum step size of 5 µm in each displacement axis. A 7-angle (−18° to 18° with a step size of 6°) compounding plane wave with the post-compounding ultrasound frame rate of 500 Hz was used to boost the contrast and ultrafast data acquisition in favor of super resolution microvessel image processing. A total of 2000 ensembles were collected, corresponding to 4 s with one bolus injection. Next, 10 V transmit voltage was selected as the most suitable voltage for MB visualization. All Time Gain Compensation (TGC) options were kept the same for all acquisitions.

### 2.2. Establishing Animal Model

All procedures involving the in vivo animal experiment were approved by the animal ethics committee of Central South University, China. A three-week-old female SD rat was purchased from Hunan SJA Laboratory Animal Co., Ltd., Changsha, China. The rat was first anesthetized via intraperitoneal injection of 3% pentobarbital sodium and two drops of levofloxacin hydrochloride eye drops were applied to the surface of the eyes. Furthermore, in order to relieve the pain during the operation, a handful of ophthalmic topical anesthetic was used. Finally, the ocular vitreum of the rat was injected with 100 thousand B16F10 cells to establish an ocular choroidal melanoma animal model [26,27].

### 2.3. Three-Frame Difference Algorithm

Figure 2 summarizes the key processing steps for the three-frame difference algorithm. Firstly, the latest seven consecutive beamformed in-phase quadrature-phase (IQ) data extracted from buffer were dealt with the absolute value approach to generate US images. These images were divided into five groups, each of which was composed of three adjacent images (N − 1, N, N + 1). In each group, two result frames containing the MB signals were then achieved via subtraction of immediately adjacent frames, while only the signals in both results that had undergone threshold processing could be retained. When the MB signals propagated through a pixel, the pixel value in binary image was set to 1 and the one-frame US image of MBs was obtained by multiplying the binary image and the US image in fame N. Finally, the real-time US image was constructed by accumulating 5 multiplied results.

### 2.4. In Vitro Tungsten Model

In order to validate the performance of the image deconvolution algorithm in the MB localization, several test experiments were completed in the custom-made module. Characterization of the point-spread function (PSF), the initial step in this study, was fulfilled to assess the diffraction limited resolution and localization precision of the US imaging system under ideal circumstances. It was challenging to extract the ideal PSFs of all MBs; therefore, the PSF of a 10 µm tungsten wire was commonly treated as an excellent alternative. The wire was held tightly between two fixings and placed in a water tank. An appropriate volume of deionized water was heated to 100 °C, and the cooled water was poured into the sink. The wire was positioned normal to the US image plane, and the transducer was moved to ensure the point scatterer phantom was imaged at the focus. The lateral and axial full width half maximum (FWHM) through the pixel was calculated for each frame. The imaging resolution of the US system was measured as the averaged FWHM of all 100 frames, and the calculated standard deviation of the centroid mean was considered as the localization precision.

When the MBs circulate with the blood flow, it is difficult for an ultrasound imaging system to distinguish the MBs that are close or adhesive, which will ultimately affect the accuracy of MB localizing and the flow velocity calculation. Therefore, we proposed the advanced deconvolution method for the super-resolution ultrasound microvessel image processing and accomplished performance verification in the self-made device. The schematic diagram of the device is shown in Figure 3a. The model consisted of two wires and a base fabricated by a high-resolution projection micro stereolithography 3D printer (microArch™ S140; Boston Micro Fabrication, Shenzhen, China). Next, UV glue was used to fix the tungsten wires on both sides of the base boss. In order to further analyze the potential priority of the deconvolution method, we selected the widths of the boss as 50 µm, 80 µm, and 100 µm, respectively. So, the distance between the center of the two wires was (N + 10) µm.

### 2.5. Flow Phantom Experiment

The flow phantom experiment was designed to confirm the superiority of the ULM algorithm and three-frame difference algorithm in selecting the optimal imaging plane in the ROI, and its schematic diagram is showed in Figure 4. The phantom was composed of an agar model and a silicone tube that had a 300 µm inner diameter and 640 µm outer diameter, and the former was made of water and agar powder in a certain proportion (98 mL water, 2 g AGAR power). As the high concentration of the injected MBs would arouse plenty signals worthy for MB localization, a suspension of sulfur-hexafluoride MBs (Sonovue, Bracco) with a concentration of 3 to 10 × 10^7^ MBs/mL that had been diluted with normal saline to approximately 1/5 times the standard concentration was used. The dual-channel syringe pump (LD-P2020II, Shanghai LANDE Medical Equipment Co., Ltd., Shanghai, China) with an adjusted flow rate was connected to the tube. In order to further estimate the capability of the three-frame difference algorithm in a large range of flow velocities, we investigated mean flow speeds of 10 mm/s, 20 mm/s, and 50 mm/s, respectively. The tube placed at the imaging plane of the US system contained a high-frequency Verasonics Vantage system (Verasonics Inc, Kirkland, WA, USA) and a L22-14v linear array probe (Verasonics Inc., Kirkland, WA, USA). In this experiment, we applied an acquisition frame rate of 500 fps and a transmit power equivalent to a mechanical index of 0.08.

### 2.6. In Vivo SD Rat Ocular Choroidal Melanoma Model

All procedures involving the in vivo animal experiment were approved by the animal ethics committee of Central South University, China. A 217 g, seven-week-old female Sprague Dawley (SD) rat was anaesthetized via isoflurane gas administered through a facial mask, followed by transferred to a heat mat, and maintained under 1.5% isoflurane during the time of the data acquisition. Two drops of phenylephrine were applied topically to prevent cornea swelling and decrease discomfort. When the imaging experiment was over, the physiological status of the mouse was constantly monitored until it regained full consciousness. To avoid the rat’s eyelids undergoing closure during the experiment, a small amount of ophthalmic topical anesthetic was used. The rat was fixed on the experimental operating table in a lateral recumbent position, and then US gel was implemented to couple the surface of the eye to the L22-14v linear array transducer (Verasonics Inc, Kirkland, WA, USA). At the beginning of the experiment, a bolus of 0.1 mL standard concentration MBs (Sonovue, Bracco) suspension was injected intravenously into the rat, and then a 1 mL flush of saline was administered. Before performing ultrasound data acquisition, in the real-time B-mode imaging with three-frame difference algorithm, the position of the rat and the ultrasound transducer were adjusted to find the optimal imaging plane in the target region. Finally, a total of 2000 ensembles were collected within 5 s and saved for offline post-processing.

### 2.7. Processing Algorithm for Super-Resolution Imaging

All data reconstruction and post-processing algorithms in this study were written in MATLAB 2020a software (The MathWorks, Natick, MA, USA). Firstly, the collected experimental data were reconstructed in Verasonics script to obtain the beamformed IQ data which were convenient for the subsequent separation of constant MBs from the background tissue. Because the computational complexity would be greatly increased when all region in the data were processed, only the ROI was selected to implement the following steps.

A spatiotemporal singular value decomposition (SVD) filter was implemented for IQ data to extract intravascular MB signals from noise and static or slowly moving tissue [28]. The total selected IQ dataset, which comprises two spatial dimensions (lateral and axial dimensions x and z, respectively) and one dimension in time (t), was reshaped to Casorati matrix by transforming time series data into a 2D space-time matrix with a dimension of (x × z, t). A SVD of the 2D Casorati matrix provided a diagonal matrix containing singular values and two unitary matrices with spatial and temporal singular vectors as columns, respectively [29,30]. The components which represent corresponding spatiotemporal coherence signals in the diagonal matrix were arranged from the most energetic to the least. It has been widely accepted that the low-order singular values illustrate the tissue signal and medium-to-high order singular values denote the signal of flowing MB [31,32]. Therefore, for the thresholds of this bandpass spatiotemporal filter, a low-order singular value threshold was determined by the turning point when the slope degree of the magnitude curve started declining to less than 45°, and the turning point was selected to be the high-order threshold as the curve flattened [33]. Then, the thresholds were adopted for an inverse SVD calculation to obtain MB signals and suppress the noise signals. As the tissue motion caused by breathing was primarily reflected in translation, rotation and scaling, a 2D phase correlation-based sub-pixel image estimation method (the “imregcorr.m” function in MATLAB) was used for motion registration [34]. When the algorithm was executed, a transformation matrix saving the estimation result was output to correct tissue motion in the MB signals. Then, intensities −20 dB or lower than the maximum value were rejected. Finally, the fine grids of images were interpolated to a pixel size of 10 µm × 10 µm through a 2D spline interpolation function, followed by a square root compression. A 2D normalized cross-correlation method which could generate cross-correlation maps was first performed between the pre-obtained PSF1 based on characterization experiment and each interpolated MB frame [24]. To further remove the remaining interference signals, a subgraph with the maximum coefficient in 2000 cross-correlation maps and the same size as PSF1 was considered as PSF2. More specially, the 2D normalized cross-correlation algorithm would be used again between the PSF2 and each MB frame to produce new cross-correlation maps, and pixels with a correlation coefficient in the map less than 0.8 were rejected. In super-resolution ultrasound microvessel imaging, MB localization is a particularly important step, which directly affects the image resolution and the performance of MBs tracking. Therefore, a Richardson–Lucy deconvolution algorithm utilizing the PSF2 as the template was applied on the image to improve the localization precision [35,36]. Next, the MB centroid was identified as the regional intensity peak value of the deconvoluted image. The details of the processing step in this section are illustrated in Figure 5.

MB tracking with excellent properties can not only exclude artifactual MBs, but also improve the accuracy of flow velocity calculation. In this study, a bipartite graph-based MB pairing method was used to pair the localized MBs between the consecutive frames, and only the reliable MB signals adapted successfully; persistence control over ten frames was retained to promote more robust tracking [37]. In view of the physiological behaviors and hemodynamic characteristics of intravascular MBs, we imposed restrictions on the velocity, direction and acceleration of MB movement to further optimize MB movement trajectories [38]. Next, a linear interpolation method was applied to fill the missing data points in each MB trajectory for the preferable visual appearance of super-resolution image. Finally, the super-resolution image of the microvessel was generated by accumulating all the interpolated images, while the flow velocity image was created by exerting the weighted spatial averaging approach and averaging the flow speeds of all the MB images [20].

## 3. Results

### 3.1. In Vitro Tungsten Model Results

Figure 3b shows an interpolated example image of the tungsten wire cross-section at the focus of the transducer. Here, the imaging system resolution was measured to be a FWHM of 194 µm in the axial and 138 µm in the lateral direction.

Figure 3c,d shows the image deconvolution results and the calculated distances of two targets with a designed distance under various numbers of iterations. We can distinctly detect that the two targets were isolated after applying the deconvolution method, and along with the increase in iteration times, the MBs localization precision was more accurate. To be specific, the 1D lateral profiles crossing the centroid peak intensity in the resulting image performed 50 iterations are displayed in Figure 3e–g. In regard to the trade-off between the computation load and clinical demands, we finally chosen appropriate 30 iterations in the subsequent data process.

### 3.2. Flow Phantom Results

Figure 6 shows the real-time MBs images processed by three-frame difference algorithm. From these images, one can see that the proposed algorithm possesses excellent performance in MB signals detection, even in a large flow speed range.

Figure 7 displays the US image, super-resolution result image and flow velocity maps of the ROI in the flow phantom. Furthermore, in Figure 7c,d, the calculated average tube diameters at the two locations were 321.8 ± 31.02 µm and 312.5 ± 29.45 µm. These results were consistent with the real tube inner diameter in the vitro experiment, indicating that our algorithm achieved outstanding performance in high-resolution imaging with fewer errors. In order to further investigate whether the super-resolution imaging algorithm could be adapted to a larger range of flow velocity scenarios, we controlled the MBs at flow rates of 10 mm/s, 20 mm/s, and 50 mm/s using a dual-channel syringe pump. As shown in Figure 7e–g, the final reconstructed averaged flow speeds were 9.6 mm/s, 20.5 mm/s, 52.0 mm/s, which is close to the pre-defined velocity.

To reveal the importance of the MB movement trajectories optimization step and image deconvolution algorithm for the ULM imaging technique, the comparison results of employing the two proposed methods are shown in Figure 8. Firstly, Figure 8a–d display the MB trajectories (Figure 8a–c) persistently tracked successfully for ten consecutive frames, and the optimized trace (Figure 8b–d) restricted by velocity, direction and acceleration. The remarkable promotion of applying the trajectory optimization step after MB tracking is clearly illustrated in these examples. Without trajectory optimization, numerous inferior trajectories presented features such as excessively long distances or disorganized or reversed paths. Figure 8e–g further manifested the capability of the image deconvolution algorithm in flow velocity calculation. The calculated velocity results of the image deconvolution algorithm were shown in Figure 8f. From these contrasting results, we can see that the calculated mean flow speed is closer to the pre-defined value and less biased after performing the algorithm. With the use of the image deconvolution algorithm, the precision of MB localization improved, and the performance of the ULM further strengthened.

### 3.3. In Vivo SD Rat Ocular Choroidal Melanoma Model

Figure 9a,b show real-time US images after three-frame difference processing at two moments. With the proposed algorithm, we can easily detect whether microvessels and the optimal imaging plane exist in the ROI. Compared with CDI, the US imaging based on the three-frame difference algorithm presents unique advantages in real-time recognition of microvascular location and morphology. In the US image (Figure 9c) of the SD rat ocular region, the eyeball with a diameter of about 6 mm and a heterogeneous echo area of 3.8 mm × 2.4 mm can be observed distinctly. Merely depending on the information provided by conventional US imaging, it is difficult to identify the region with indistinct borders as OCM. Furthermore, its clinical similarity to benign choroidal nevus also greatly limits the early diagnosis of OCM. However, in the corresponding super-resolution microvessel image superimposed on the original US image (Figure 9d), the structure and distribution of the vasculature network are clearly visible in fine scale. The microvessels less than 1 mm in the OCM are clearly indicated in the zoom-in view of super-resolution imaging in Figure 9e. The super-resolution microvessel blood flow speed map was shown in Figure 9f. The estimated averaged velocity was 12.23 mm/s, which fit well with the expected values.

Figure 10a exhibits the advanced image of PSF2 filtered via the first 2D normalized cross-correlation method. Its resolution was measured to be a FWHM of 194 µm in the axial and 138 µm in the lateral direction. Besides, the peak intensity value was 3356, which was five times that of PSF1. According to the comparison results of the two PSFs, we can see that the PSF2 derived from in vivo SD rat data has a higher resolution, and its peak intensity is closer to the actual experimental conditions, which is beneficial for precisely extracting microbubble signals and effectively suppressing background noise. It is evident that the optimized PSF2 more accurately conforms to the definitions of PSF in ULM technology. The 1D cross-section profiles of three-line markers (as indicated in Figure 9e) are depicted in Figure 10b–d. Not only can we detect microvessels with a FWHM ranging from 90 µm to 240 µm, but we can also separate the microvessels to 290 µm apart.

## 4. Discussion

OCM is a familiar malignant tumor, which is characterized by high metastatic risk and poor prognosis. It threatens the life of patients, and its early detection would be of great clinical significance. Estimating the blood flow velocity and microvessel morphology inside the tumor in vivo can provide significant information for its early clinical diagnosis, treatment decision making and better prognosis. Currently, B-mode US imaging and CDFI are widely accepted imaging modalities for OCM diagnosis. However, the small OCM in the early stage commonly presents tiny microvessels and the overlapping clinical features with the benign choroidal nevus. These factors have greatly limited the performance of B-mode US imaging and CDFI. ICGA and CT, the high spatial resolution technologies used to assess the deep ocular vasculature, will also limited by refractive medium opacities. Thus, we proposed an advanced ULM technique based on an image deconvolution algorithm used to visualize the microvessel morphological features of early OCM at high resolution and precisely estimate the blood flow velocity. The technology could be complementary to diagnosis of OCM and could provide patients with a better prognosis, which should not be neglected.

Before implementing data acquisition, positioning the probe and selecting an optimal imaging plane were critical for this present study. However, the step was performed only with the assistance of the conventional B-mode and color Doppler, which limited the further popularization of the ULM technique for clinical application. Therefore, we propose real-time US plane wave imaging based on the three-frame difference algorithm to capture MBs signals. After the MBs were injected, according to the real-time imaging results, the position of the probe was adjusted to select an optimal imaging plane. In contrast with the guidance of B-mode and color Doppler, our proposed method enables clearer and direct observation of the microvascular network and is not restricted by the vessel diameter and blood flow speed. The results in flow phantom and vivo (Figure 6 and Figure 9a,b) demonstrate the efficacy of the real-time imaging based on the three-frame difference algorithm.

ULM reconstructs the super-resolution image of the microvasculature by accumulating the centroid position of intravascular isolated MBs. One of the essential aspects is the capacity to detect the centroid of MBs with high accuracy in a finer grid [39]. Nonetheless, in most previous publications regarding super-resolution imaging, the intensity peak of MB signals was directly used to define its centroid position, which might lead to hundreds of micrometers in error [40]. Additionally, in the theory of US imaging, the image is considered to be a convolution of the system PSF with a point source. Hence, utilizing the image deconvolution in the MB localization process could be more coincident with the reality.

In this study, we first performed a tungsten model experiment in the custom-made module to verify the improvement of MB localization precision through the image deconvolution algorithm. From Figure 3c,d, one can see that the measured distance between two target points grows closer to the pre-defined value with the increasing iteration times. In particular, when the distance was set to 50 µm, the result before image deconvolution processing could be recognized as a point, which would probably not be a negligible error for super-resolution imaging. Next, we compared the velocity estimation results (Figure 8e–g) in the flow phantom model before and after deconvolution processing. Despite the large range of test velocity, the reconstructed results using the image deconvolution algorithm presented a better performance and were less subjected to quantization error.

The results of the in vivo animal study also confirmed the feasibility of ULM in guiding early stage small OCM diagnosis. In the ultrasound image (Figure 9c), we could only observe a heterogeneous echo area of 3.8 mm × 2.4 mm on the right side of the eye. However, in order to further identify the small suspicious choroidal lesions, the patient was required to undergo a period of regular monitoring. Observing the tumor growth and changes to differentiate OCM from benign choroidal nevus could cause patients to suffer from greater risk of metastasis. Currently, the guidance of ULM technology can be salutary for improving clinical diagnosis precision, and it may assist therapeutic decision making for patients.

There are a few limitations of this study. First, in view of the differences in OCM growth, morphologic and molecular characteristics between human and SD rat models, more clinical data collected from patients are indispensable to determine the clinical applicability of ULM. Secondly, although the ULM showed an admirable performance in in vitro flow phantom tests, many factors can influence the validation of blood flow velocity measurement in vivo, including complicated physiological structure, heterogeneous acoustic medium and unpredictable blood flow speed. Third, the lack of optical comparison and the use of only one animal model further limited the possibility of the ULM being used in clinical application. Finally, only one pre-defined PSF was used in the 2D cross-correlation algorithm and image deconvolution algorithm. While a new method obtaining PSF was proposed in this paper to optimize the robustness of ULM, the MBs flow with the blood in different directions and the ULM can perform a 2D scanning strategy constrained by the linear array transducer. This will lead to false detection and localization of MB signals, thus affecting the super-resolution imaging of microvasculature. In the future, the most significant investigation direction will comprehensively combine various imaging technologies to further improve the clinical diagnosis and prognosis judgement of OCM.

## 5. Conclusions

In summary, our study demonstrated a B-mode image, super-resolution image and flow speed map of OCM, providing a supplementary approach to the differentiation of small malignant OCM and benign choroidal nevus. ULM and real-time imaging based on the three-frame difference algorithm was successfully performed in the OCM, which may promote the application of ULM to early detection of more clinical diseases. We demonstrated the image deconvolution algorithm’s awesome capacity to suppress the background signals and noise, facilitating more robust MB localization and precisely calculating the flow velocity. ULM can detect the microvascular distribution and blood supply in OCM with high resolution to help clinicians evaluate the progression and grade of OCM and will be a promising way to implement the translation from preclinical to clinical.

## Figures and Tables

**Figure 1 bioengineering-10-00428-f001:**
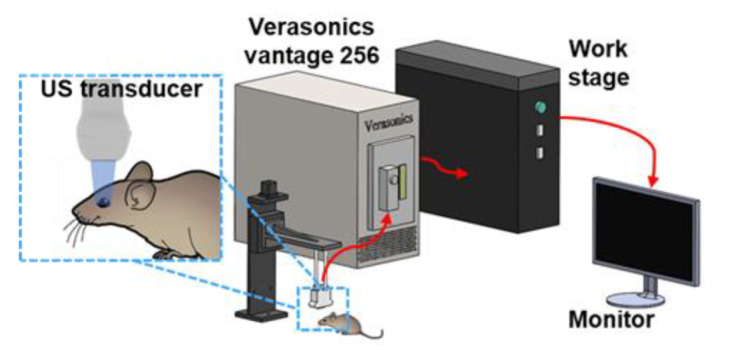
Schematic diagram of the ULM for microvessel imaging of the OCM.

**Figure 2 bioengineering-10-00428-f002:**
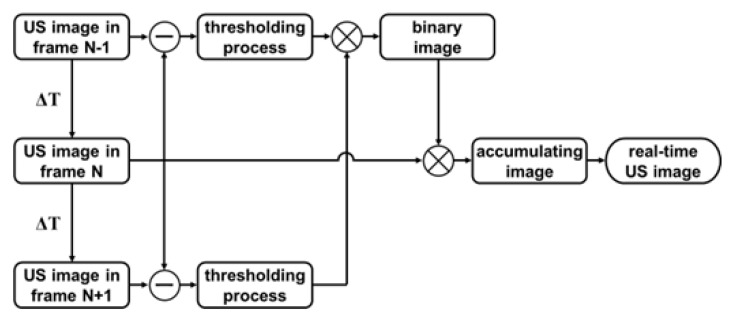
Flow chart of the three-difference algorithm for generating real-time US image.

**Figure 3 bioengineering-10-00428-f003:**
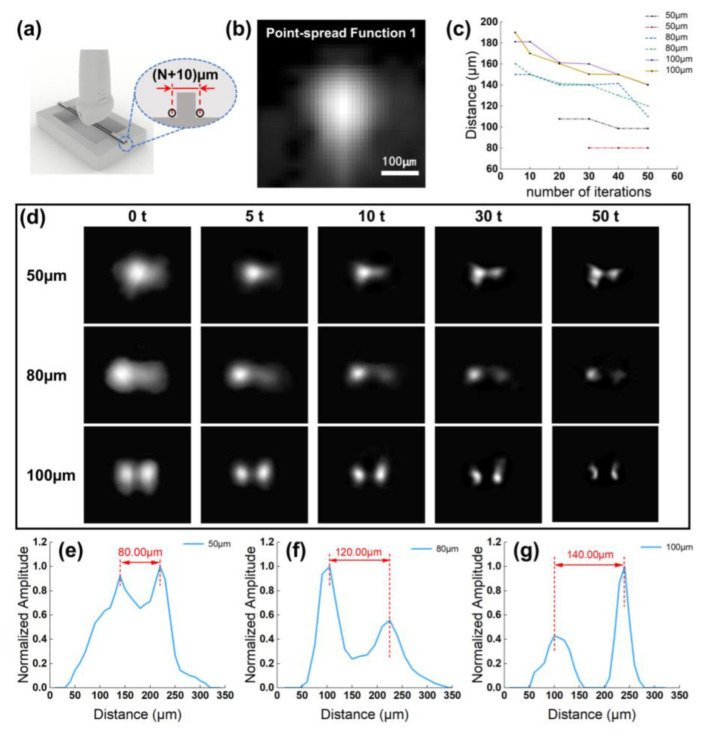
The quantitative analysis for image deconvolution algorithm in vitro tungsten model experiment. (**a**) Schematic diagram of the custom-made tungsten wire model. (**b**) Example US image of 10 μm tungsten wire target cross section at the focus of the imaging system. (**c**) The performance of Richardson–Lucy deconvolution algorithm in improving MBs localization precision. The axes indicate the calculated distance between two intensity peaks on the image after image deconvolution processing. Two sets of data were acquired for each pre-defined value. (**d**) US images of two-point scatterer phantom under different number of iterations in image deconvolution algorithm. From the 1st row to 3rd row, the distance between the center of the two points are 50 μm, 80 μm, 100 μm. Each column represents a different number of iterations. (**e**–**g**) The 1D lateral profiles crossing the centroid peak intensity corresponds to the last column of (**d**).

**Figure 4 bioengineering-10-00428-f004:**
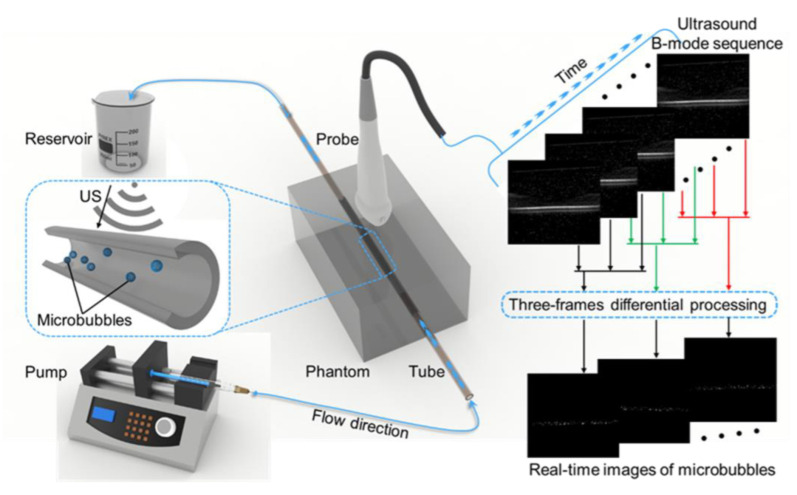
Schematic diagram of the ULM for imaging the phantom. The right side of the diagram shows the reconstructed phantom US images and the real-time MB images processed by the three-frame difference algorithm.

**Figure 5 bioengineering-10-00428-f005:**
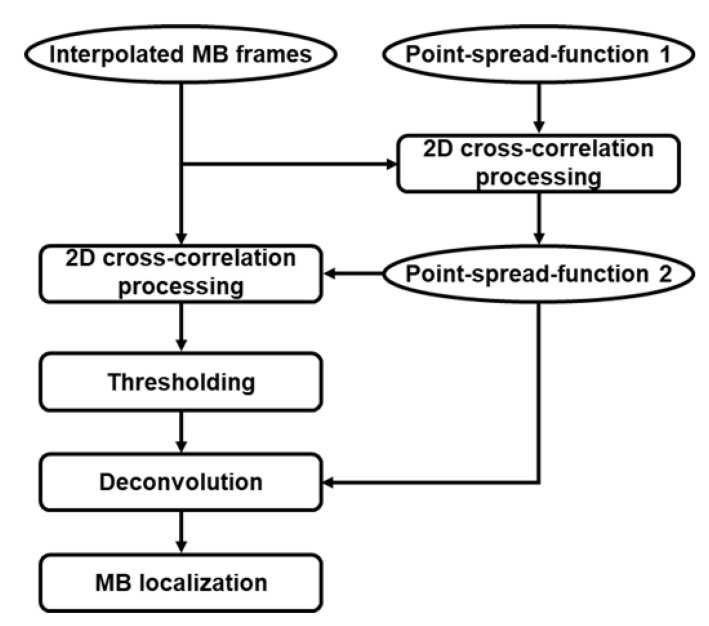
The part of post-processing chain for super-resolution ULM image.

**Figure 6 bioengineering-10-00428-f006:**
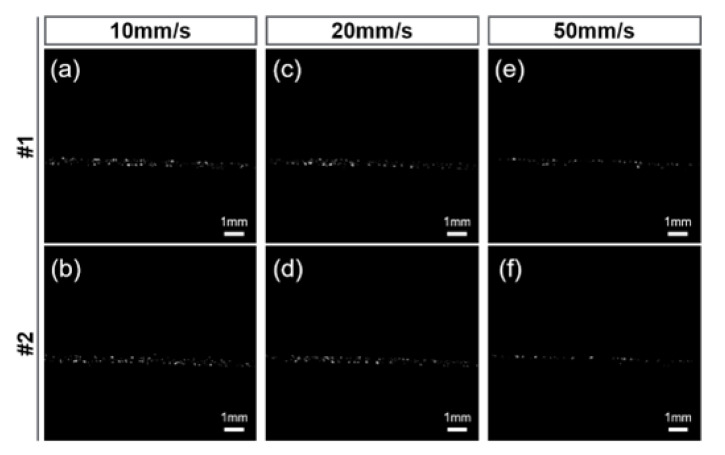
The examples of real-time B-mode imaging with three-frame difference algorithm under different flow velocities. Images at two different moments (#1, #2) were listed for each set of flow velocities. (**a**,**b**) 10 mm/s, (**c**,**d**) 20 mm/s and (**e**,**f**) 50 mm/s.

**Figure 7 bioengineering-10-00428-f007:**
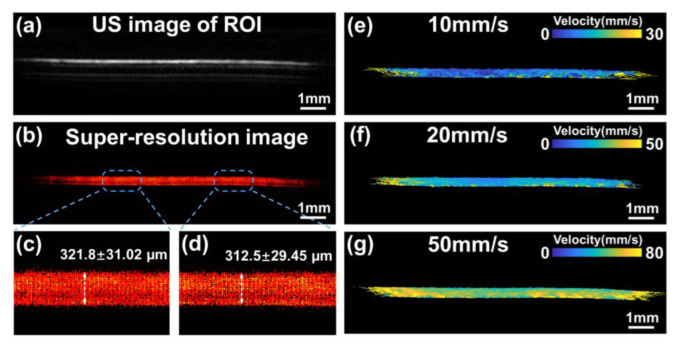
(**a**,**b**) The US image and the super-resolution image of the flow phantom. (**c**,**d**) The zoom-in views of the box regions in (**b**). (**e**–**g**) The reconstructed flow velocity map at 10 mm/s, 20 mm/s, 50 mm/s.

**Figure 8 bioengineering-10-00428-f008:**
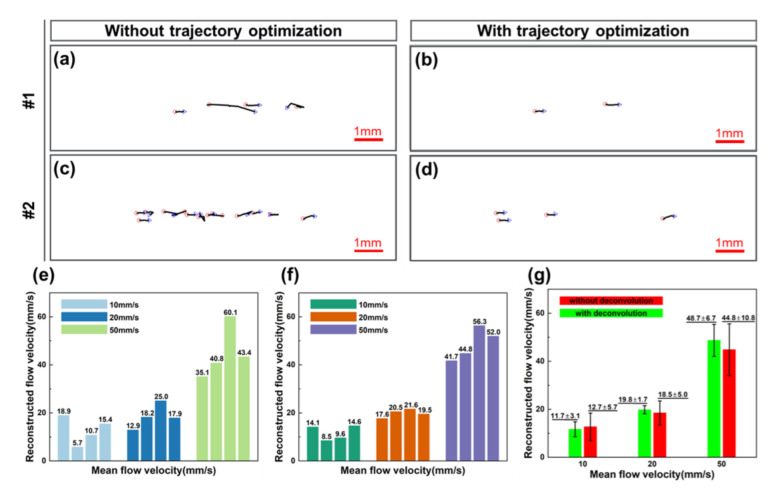
(**a**–**d**) Comparison of MB trajectories persistently tracked successfully for consecutive ten frames before and after optimization in flow phantom. (**e**–**g**) The bar plot of calculation results, mean and standard deviation under different test speeds before and after image deconvolution processing.

**Figure 9 bioengineering-10-00428-f009:**
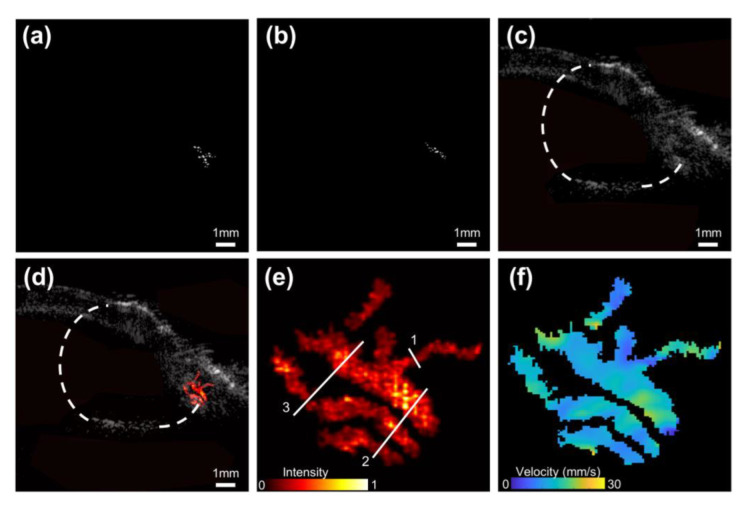
(**a**,**b**) Examples of the real-time US images after three-frame difference processing under different moments. (**c**,**d**) Examples of the real-time conventional B-mode image and its superimposed image with the reconstructed super-resolution microvessel image in vivo. Eyeball region is marked in white dotted lines. (**e**,**f**) Super-resolution microvessel density map and flow speed map inside OCM of SD rat. Three-line markers in (**e**) were used to calculate the detectable resolution and the resolved distance.

**Figure 10 bioengineering-10-00428-f010:**
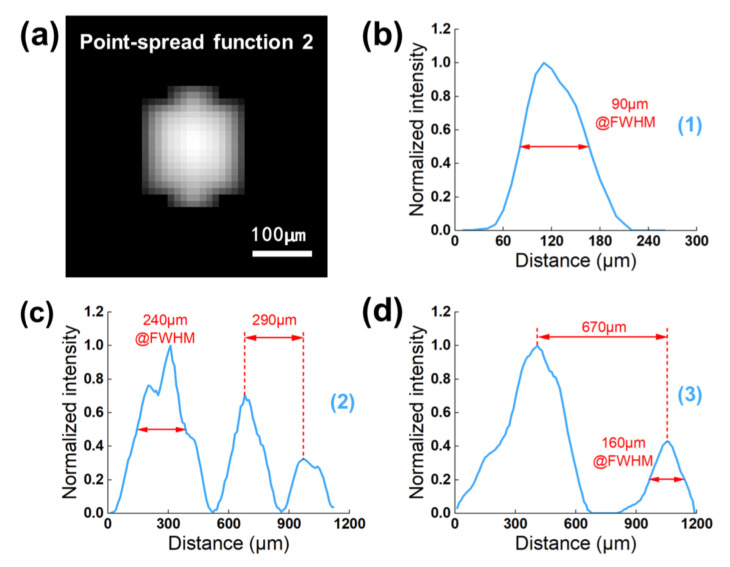
(**a**) The result image of PSF2 after first performing the 2D cross-correlation method. (**b**–**d**) 1D cross-section profiles of three line markers indicated in Figure 9e.

## Data Availability

The data presented in this study are available on request from the corresponding author. The data are not publicly available due to the confidentiality of the relevant codes.

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
