# Peer review of "Detecting Early Ocular Choroidal Melanoma Using Ultrasound Localization Microscopy"

_bioengineering, 2023, doi:10.3390/bioengineering10040428_

Round 1

Reviewer 1 Report

"The study, titled "Detecting Early Ocular Choroidal Melanoma Using Ultrasound Localization Microscopy" presents an ultrafast plane wave imaging technique that captures MBs in ROI based on a three-frame difference algorithm and completes the detection of OCM in vivo rat using ULM. I recommend the publication of this paper in the bioengineering after making minor revisions.

The authors have revised their manuscript based on the following questions:

1.The authors have emphasized the importance of US microbubble discussion in the introduction part.

2. The quality of all the figures should be improve.

3. As suggested, the entire manuscript for minor spelling and text corrections are required.

Reviewer 2 Report

In this paper, the authors propose a novel method for obtaining the kernel of the image deconvolution algorithm and present an ultrafast plane wave imaging that capturing method based on a three-frame difference algorithm. By utilizing these new methods. The authors improve the performance ULM greatly and fulfill the detection of OCM in vivo rats with ULM.

The experiment is well-designed and discussed thoroughly. 

 There are some minor improvements that should be addressed before publication.

1. Figure 5 is missing in the manuscript.

2. what is the difference between #1 and  #2 in figure 6? It is better to clarify their meaning in the caption.

3. Does figure 9a show a real-time US image, while 9b gives the one after three-frame difference processing? The description in the caption is not clear.

Reviewer 3 Report

Quan et al. evaluated a method of ultrasound imaging applying the 3-frame algorithm for improved visualization of choroidal melanoma. Although the authors introduce a method that may potentially facilitate an early diagnosis, no comparison to the state-of-the-art OCT or OCTA system was provided, so it’s still an open question whether this approach improves the differential diagnosis. Besides, other points must be addressed before this manuscript can be considered for publication.

1.      Abbreviations, such as OCM or US, need to be spelled out in the main text not only in the abstract.

2.      L110-11 – a reference is missing on the effectiveness of the applied procedures to establish the animal model.

3.      L200 – what was the ROI size and its selection procedure?

4.      Fig. 5 is missing.

5.      Fig. 6 – how does it compare to the conventional method?

6.      L310-11 – how can “the optimal imaging plane in the ROI” be found, and how can the 3-frame algorithm facilitate this process?

7.      L315-18 – please point to parts of fig. 9 you refer to in these two statements.

8.      Fig. 9 – please explain the difference between (a) and (b)

9.      Fig. 9 - in clinical decision-making it’s important to determine the height and diameter of melanoma. Can the authors provide such parameters?

10.   Reproducibility is essential in follow-up examinations. Did the author look at other time points?  

11.    Fig. 9e – please add a scale bar.

12.   L374 – ‘faculty to detect’??

13.   L399 – the comparison to optical methods and the use of only one animal model also appear to be the limitations of the current work.  

14.   The manuscript contains numerous expressions that are uncommon in scientific writing, for example, tremendous, splendid, admirable, etc.
